# The Devil is in the EOS:
# Sequence Training for Detailed Image Captioning

**Abdelrahman Mohamed    Yova Kementchedjhieva**
Mohamed bin Zayed University of Artificial Intelligence (MBZUAI)
{abdelrahman.mohamed, yova.kementchedjhieva}@mbzuai.ac.ae

## Abstract

Despite significant advances in vision-language models (VLMs), image captioning often suffers from a lack of detail, with base models producing short, generic captions. This limitation persists even though VLMs are equipped with strong vision and language backbones. While supervised data and complex reward functions have been proposed to improve detailed image captioning, we identify a simpler underlying issue: a bias towards the end-of-sequence (EOS) token, which is introduced during cross-entropy training. We propose an unsupervised method to debias the model's tendency to predict the EOS token prematurely. By reducing this bias, we encourage the generation of longer, more detailed captions without the need for intricate reward functions or supervision. Our approach is straightforward, effective, and easily applicable to any pretrained model. We demonstrate its effectiveness through experiments with three VLMs and on three detailed captioning benchmarks. Our results show a substantial increase in caption length and relevant details, albeit with an expected increase in the rate of hallucinations.

## 1 Introduction

As vision-language models (VLMs) advance, new frontiers open up in multimodal research, some examples being the novel tasks of interactive visual dialogue (Liu et al., 2023) and culturally-aware visual question answering (Romero et al., 2024). The fundamental task of image captioning (Vinyals et al., 2015) has also seen an upgrade: in place of the classic benchmark MS COCO (Lin et al., 2014), a number of new evaluation datasets have been proposed, consisting of images with detailed (a.k.a fine-grained or dense) captions (Onoe et al., 2024; Cho et al., 2022; Urbanek et al., 2024). Detailed image captioning has extrinsic applications, e.g., assistive tools, as well as a key intrinsic application, as a form of image textualization for solving knowledge- and reasoning-intensive vision-language tasks using large language models (Pi et al., 2024).

At its core, detailed image captioning does not require any new capabilities compared to coarse image captioning, i.e. both forms of the task rely on the ability of the vision encoder in a VLM to extract useful features from the input image, and of the LM backbone to decode the semantics of these features and express them in coherent text. Typically, the vision encoder and the language decoder in a VLM are initialized from extensively pretrained models, capable of complex feature extraction and long-form text generation, respectively. As such, detailed image captioning should be well within the reach of these components. Yet, during joint vision-language training the language decoder is trained to match the distribution of short coarse captions, thus losing the ability to generate long-form text.

Various methods have been considered for training VLMs to generate detailed captions. State-of-the-art large VLMs are being extensively trained on instruction-tuning data for a range of tasks, including detailed image captioning (Liu et al., 2023). Smaller and more practical VLMs have also been trained specifically for this task, often using synthetic data induced from a larger VLM (OpenAI, 2023). Away from direct supervision, some have experimented with reinforcement learning using a teacher VLM as a reward model (Cho

et al., 2022), or sourcing distant supervision from downstream tasks (Fisch et al., 2020). In this work, we introduce a method for adapting pretrained VLMs to perform detailed image captioning *without any form of supervision*.

We view the generation of detailed captions as a form of recovering existing model capabilities (rather than instilling new ones) and propose to achieve that through end-of-sequence (EOS) token debiasing. The EOS token exists in every single caption in the training data; combined with teacher forcing, this causes the model to have a bias towards predicting it. Alleviating this bias should lead the model to generate longer text (Newman et al., 2020; Kulikov et al., 2021), and the conditioning on the image context during vision-language pretraining should ensure that this longer text remains grounded in the input image. Through our experiments, we show that simply pushing down the probability of the EOS token through sequence training of models trained on generic captions leads to longer, more detailed and largely accurate outputs.

We conduct experiments on three popular VLMs: two variants of BLIP-2 (Li et al., 2023) and PaliGemma (Beyer et al., 2024), chosen as they have not been explicitly trained for the task of detailed image captioning. Our finetuned models are evaluated against the base models on three dense captioning evaluation datasets. We find that across different models and metrics, our approach results in a substantial increase in the length of generated captions. Through a trade-off analysis of recall and hallucination metrics, we determine that the additional text in the caption provides relevant details, proving the effectiveness of sequence training with EOS debiasing as a way to induce detailed captions from pretrained VLMs.

## 2 Related work

### 2.1 Vision Language Modeling

Here we introduce the common vision-language modeling framework as exemplified in the two VLM families included in our study: BLIP-2 (Li et al., 2023) and PaliGemma (Beyer et al., 2024).

Current approaches to vision-language modeling adopt an encoder-decoder approach, wherein input images are encoded into visual tokens, which are subsequently decoded into text. Both components are typically initialized with pretrained weights and exhibit powerful generalization capabilities.

In order to learn a mapping of visual features into language semantics, the VLM is trained on image-text pairs. BLIP-2 is pretrained on image-caption pairs, and PaliGemma on a mixture of tasks: captioning, question-answering, OCR, object detection, and segmentation.[1] While these multimodal pretraining datasets are very large indeed, they amount to a small fraction of the data used to train the language backbone of VLMs, and represent a considerably narrower distribution of mostly short texts. As such, the multimodal pretraining phase helps the language decoder to learn how to interpret visual features and map them into language, but it also shifts the distribution of the underlying language model in line with the text found in the multimodal training data.

### 2.2 Detailed Image Captioning

Several works have attempted to improve the ability of VLMs to generate detailed image captions with fully supervised and weakly supervised approaches. We discuss these methods in terms of the type of supervision and training objectives they employ.

**Fully Supervised Methods** rely on training datasets of images paired with detailed captions. The captions in these datasets are either human annotated (Urbanek et al., 2024; Onoe et al., 2024), with high annotation costs limiting the amount of data produced, or synthetically generated, using large closed-source VLMs (Chen et al., 2023; Singla et al.,

---

[1]Note that this mixture of tasks does not include detailed image captioning: the data used for the captioning task consists of alt-text scraped from the web.

2024; Gaur et al., 2024). Recent work shows increased hallucination and bias in models trained on synthetic captions (Hirota et al., 2024).

**Weakly Supervised Methods** use no explicit training data for detailed image captioning, and instead leverage sequence training (Ranzato et al., 2016) with reward signal from an external component. Fisch et al. (2020) use a question-answering system as a reward model: generated captions are evaluated based on whether questions about the input image can be correctly answered with reference just to the caption. Luo et al. (2018) introduce a discriminability objective to encourage captions that uniquely describe one image and cannot be equally applied to another similar image. They use the success of a text-to-image retrieval model as a reward. Cho et al. (2022) use CLIP as a reward model to compute a similarity score between the generated caption and the input image, with the intuition that more detailed captions would yield higher scores. In all these works, there is a high cost associated with the reward model and potential for reward hacking through shortcut solutions.

Closest to our work is the SMILE method (Yue et al., 2023), which trains on short captions using the next-token prediction objective, but modifies the Maximum Likelihood Estimation step to penalize missing tokens from the ground-truth captions, but not any extra ones that the model adds. The model can thus generate more detail but (a) still has to adhere to the word choice and syntactic framing of the ground-truth caption, and (b) does not have an incentive to actually increase the caption length.

Our work proposes a method based on sequence training, which alleviates the issue of exposure bias (Rennie et al., 2017), allows the model more freedom in word choice (see §3.1), and targets caption length explicitly, through suppression of the EOS token.

### 2.3 The EOS Problem

Language generation models are typically trained with cross-entropy loss and teacher forcing (Williams & Zipser, 1989), using an EOS token to signal completion. Newman et al. (2020) found that the EOS token plays a critical role in the model's ability to extrapolate to sequence lengths not encountered during training. Their experiments revealed that training without the EOS token allowed a Dyck-(k,m) language model (Chomsky & Schützenberger, 1959) to generalize to sequences up to ten times longer than those seen in the training data. This behavior was attributed to the EOS token introducing an implicit counter that tracks the sequence length, and adhere to the length observed during training.

Kulikov et al. (2021) observed that in machine translation, treating the EOS token like any other token leads to an oversmoothing problem. This results in the model assigning high probabilities to short sequences, leading to premature termination of the generation process. In contrast, Yue et al. (2024) studied instruction-finetuned VLMs, noting that these models, trained on longer captions, often over-generate, failing to terminate even when all relevant information has been covered. This issue, in part, arises because the EOS token probability is weakened, preventing the model from properly signaling the end of the sequence. These findings underscore the importance of the EOS token in striking a balance between under- and overgeneration.

## 3 Approach

Our approach addresses the bias towards early EOS token generation caused by supervised training on short captions. This is achieved by gradual debiasing of the EOS token through sequence training. We begin with an introduction of sequence training.

### 3.1 Sequence Training for Image Captioning

Ranzato et al. (2016) introduce sequence training as a way to mitigate exposure bias, a problem that arises when a language model, trained with teacher-forcing, accumulates

errors during inference. This occurs because, during inference, each new token is generated based on previously generated tokens, rather than the ground-truth tokens used in training.

Rennie et al. (2017) adapt sequence training to image captioning, replacing the traditional next-token prediction objective with direct optimization of an evaluation metric applied to complete sequences (captions) generated by the model. Given an input image $I$, a sequence of tokens is generated, $S = [t_1, t_2, ...., t_n]$ where $t_n$ is the EOS token. Next, a reward, $R(S)$, is calculated against the reference captions for $I$ using the CIDEr metric (Vedantam et al., 2015). Finally, the gradient is computed using REINFORCE (Williams & Zipser, 1989):

$$\nabla \mathcal{L}(\theta) = -\frac{1}{n} \sum_{i=1}^{n} R(S) * \nabla_\theta \log p_\theta(t_i) \tag{1}$$

where $\nabla_\theta \log p_\theta(t_i)$ is the gradient of the $i^{th}$ predicted token. As the equation above iterates through $i = \{1, ..., n\}$, all tokens in the generated sequence are rewarded or penalized uniformly.

## 3.2 EOS Debiasing

Our goal is to train VLMs to generate detailed image captions using the benefits of sequence training without the cost of an external reward model (Fisch et al., 2020; Luo et al., 2018; Cho et al., 2022). Given the observations on the critical role of the EOS token in sequence length generalization (Newman et al., 2020), we hypothesize that by manipulating this token alone, we should be able to induce longer generated text. Concretely, we propose to gradually reduce the probability of the EOS token through sequence training. For each generated caption in a training batch, we minimize the probability of EOS token:

$$\nabla \mathcal{L}(\theta) = \nabla_\theta \log p_\theta(t_n) \tag{2}$$

Where $t_n$ is the EOS token. The reasonable expectation here is that this simple zero-cost signal will suppress the tendency learned during vision-language pretraining to end the sequence early. The purely empirical question, however, is what strategy the VLM would adopt in response to this intervention.

## 3.3 Prerequisites

This approach consists of negative feedback that forces the model into longer generation, without positive reinforcement to guide the VLM towards what *should* be generated. It instead relies on the assumption that the VLM will find a solution consistent with its prior training, i.e. that the new longer text generated by the VLM will adhere to the rules of language learned during the text-only pretraining of the language backbone, and to the visual-token conditioning learned during the vision-language pretraining stage. As such, the approach of EOS debiasing relies on the strong fundamental capabilities of VLMs, and similarly to other sequence training methods (and SMILE), it cannot be applied during the initial stage of VLM pretraining.

## 4 Experiments

We test the proposed approach on three VLMs, built on strong language backbones and finetuned for the task of image captioning: BLIP-2 with an OPT 2.7B decoder,[2] BLIP-2 with a FlanT5-XL decoder,[3] and PaliGemma 3B[4]. We choose the checkpoints finetuned on the MS COCO dataset (Lin et al., 2014) and adopt the same dataset in our experiments, as this ensures a good distribution alignment between the initial finetuning and our sequence training, and allows us to quantify exactly how much extra detail we can induce through

---

[2]`Salesforce/BLIP-2-opt-2.7b-coco`

[3]`Salesforce/BLIP-2-flan-t5-xl-coco`

[4]`google/paligemma-3b-ft-cococap-224`

| Dataset | # Imgs | # Caps | Len | Description |
|---------|--------|--------|-----|-------------|
| FineCap | 1,000 | 5 | 26 | Images from COCO and Conceptual Caption with captions detailing background, objects, attributes, and spatial relations. |
| DCI | 7,805 | 10 | 45 | Images from SA-1B with detailed captions of up to 77 tokens, summarized with an LLM from hyper-detailed captions of >1k tokens. |
| DOCCI | 5,000 | 1 | 136 | Original images with captions detailing objects, attributes, spatial relations, text rendering, world knowledge, and scene setting. |
| Urban-1k | 1,000 | 1 | 101 | Images of urban scenes sampled from Visual Genome with synthetic captions detailing objects, attributes and spatial relations. |

Table 1: Summary of the evaluation datasets in terms of number of images we use, number of captions available per image, average length (in words) per caption, and other details.

EOS debiasing for the same images that the base model was trained on. The full recipe we propose essentially amounts to two-stage training on the same data: finetuning with the standard next-token prediction objective, and sequence training with EOS debiasing to generalize beyond the length learned in the first stage.

### 4.1 Implementation details

We finetune only the cross-modal bridge of the VLM (Q-former and linear projection in BLIP-2, linear layer in PaliGemma). The rest of the model parameters are frozen. The learning rate is set to $1e$-7, to allow the VLM to gradually find a solution to the longer generation task. We find that higher values lead to degenerate model behavior, while lower values excessively delay model convergence. Learning rate ablation can be found in A.3.2. Models are trained with a batch size of 8 and 3 gradient accumulation steps on an NVIDIA A100 80GB card. We use contrastive decoding (Su & Collier, 2022) during training, to encourage exploration through more diverse outputs. We train until the generated captions reach a sequence length of 60, the maximum that fits on this GPU card. All generation is done using beam search decoding with 5 beams, a repetition penalty of 1.5 and a no-repeat-ngram constraint of 3.

### 4.2 Evaluation Datasets

We evaluate the proposed approach on three datasets for detailed image captioning: FineCapEval (Cho et al., 2022), DOCCI (Onoe et al., 2024) and DCI (Urbanek et al., 2024). Furthermore, we measure retrieval performance on the Urban-1k dataset (Zhang et al., 2024); see Table 1 for details.

### 4.3 Metrics

**Performance metrics** We measure performance using two established metrics: CIDEr (Vedantam et al., 2015), and the recently introduced CAPTURE (Dong et al., 2024). Additionally, we measure the level of coherence of the generated captions, using a large language model as a judge (Zheng et al., 2023). CIDEr is a widely used metric in traditional image captioning but, as we find, highly unreliable in the context of detailed image captioning. CAPTURE utilizes a text scene graph parser to extract visual elements such as objects, attributes, and relations from both the predicted and reference captions. Then it applies hard and soft matching to these extracted elements and computes an F1 score. As such, CAPTURE indicates how accurate and complete the caption is, but not how well-formed it is. We therefore utilize GPT-4 to measure coherence: for each caption, we prompt the model to evaluate the coherence on a scale of 1 to 5 following the evaluation criteria shown in Appendix A.4.

| Model | | FineCapEval | | | | DCI | | | | DOCCI | | | |
|---|---|---|---|---|---|---|---|---|---|---|---|---|---|
| | | CIDEr ↑ | Coh ↑ | CAPT ↑ | Len | CIDEr ↑ | Coh ↑ | CAPT ↑ | Len | CIDEr ↑ | Coh ↑ | CAPT ↑ | Len |
| BLIP-2 OPT | base | 21.86 | **3.12** | 38.92 | 9.57 | 1.67 | **3.10** | 35.42 | 11.98 | 0.00 | **3.24** | 32.44 | 11.58 |
| | triv. | 14.32 | 2.05 | **46.72** | **35.79** | 10.71 | 2.14 | 43.51 | **41.64** | 0.00 | 2.15 | 42.13 | **40.51** |
| | ours | **24.87** | 2.87 | 46.31 | 28.69 | **11.98** | 2.41 | **44.23** | 33.95 | 0.00 | 2.54 | **45.23** | 31.44 |
| BLIP-2 T5 | base | 16.93 | 3.13 | 37.32 | 9.56 | 1.21 | 3.03 | 34.62 | 10.05 | 0.00 | 2.91 | 30.85 | 9.87 |
| | triv. | **17.72** | 1.33 | 40.91 | 30.41 | 5.23 | 2.17 | 40.91 | 31.13 | 0.00 | 1.37 | 37.59 | 31.93 |
| | ours | 12.82 | **3.27** | **46.52** | **34.11** | 8.72 | **3.51** | **46.47** | **38.99** | 0.51 | **3.45** | **47.13** | **36.56** |
| Pali Gemma | base | 17.91 | **3.59** | 39.82 | 10.12 | 0.91 | **3.42** | 35.02 | 10.29 | 0.00 | **3.57** | 31.00 | 10.11 |
| | triv. | 11.91 | 1.89 | 43.08 | **32.84** | 4.12 | 2.02 | 40.73 | **33.72** | 0.00 | 1.37 | 39.98 | **32.25** |
| | ours | **38.01** | 3.25 | **48.06** | 21.12 | **5.41** | 3.41 | **43.84** | 22.83 | **0.01** | 3.23 | **41.53** | 21.72 |
| LLaVA | | 3.62 | 4.37 | 54.41 | 70.56 | 6.54 | 4.55 | 53.21 | 76.81 | 2.23 | 4.53 | 56.31 | 61.36 |

Table 2: Comparison between base model, trivial solution, and finetuned models (in orange) on descriptive captioning datasets. LLaVA results (Tiny-LLaVA) included as SOTA reference. Our models consistently outperform the baselines. Best number in each section is bolded.

Following Luo et al. (2018), we also evaluate the predictions of our models in a image-text retrieval setting. We obtain image and text embeddings from LongCLIP (Urbanek et al., 2024), a vision-language representation model designed to handle long text. Candidate ranking is based on cosine similarity and performance is measured in terms of Recall@1.

**Hallucination metrics** Since hallucination is a known and expected problem in detailed image captioning (Zhou et al., 2024b), we also include two hallucination metrics: $CHAIR_i$ (Rohrbach et al., 2018) and ALOHa (Petryk et al., 2024). $CHAIR_i$ counts how many extra objects are mentioned in the generated caption compared to the reference captions, i.e. how many objects are hallucinated. Its complement, $Recall_i$, counts how many of the objects in the reference captions are mentioned in the generated caption. The two metrics are constrained to the COCO vocabulary of 80 object classes and as such are only reasonably applicable to the COCO dataset. ALOHa is an open vocabulary method which utilizes a large language model to extract visually grounded objects from a generated caption and then matches them to objects derived from reference captions and extracted from the input image using an object detector. We report unigram Recall alongside ALOHa as a proxy for caption completeness, by analogy to the $CHAIR_i$–$Recall_i$ pairing.

**Baselines** We compare the models we train first and foremost against the corresponding base VLMs, in order to demonstrate the relative improvement in detailed caption quality. As a trivial alternative to EOS debiasing through sequence training, we perform EOS blocking at inference time, implemented using the `bad_words_ids` argument in the `generate()` function in `transformers` (Wolf et al., 2020). Effect of decoding strategy in the trivial baseline can be found in A.3.1. To contextualize our models' performance, we compare against TinyLLaVA (Zhou et al., 2024a) a recent 3B parameters VLM explicitly trained for detailed image captioning (among other instruction-tuning tasks), which surpasses larger models, such as LLaVA 7B (Liu et al., 2023).

## 4.4 Main Results

**Image Captioning** results are shown in Table 2. Firstly, we observe a failure mode of the CIDEr metric: while the numbers for the relatively short FineCapEval dataset appear to be in a reasonable range, they drop substantially for DCI and collapse for DOCCI. It appears that with the growing number of n-grams in the reference captions across DCI and DOCCI, the chance for a mismatch is higher, leading to progressively lower results, to the point that the metric becomes meaningless.

Next, we focus on CAPTURE. In line with the design of this metric, tailored specifically to detailed image captioning, we see that scores remain stable across the three datasets. Compared to the base model, the trivial baseline of EOS token blocking at inference time offers a consistent improvement. This is a strong baseline, since beam search guides the

| Model | | Urban-1k | | COCO | | DCI | | Len ↑ |
|---|---|---|---|---|---|---|---|---|
| | | Txt2Img ↑ | Img2Txt ↑ | Recall$_i$ ↑ | CHAIR$_i$ ↓ | Recall ↑ | ALOHa ↑ | |
| BLIP-2 OPT | base | 28.43 | 35.71 | 48.6 | **2.5** | 17.1 | **0.556** | 11.61 |
| | ours | **49.42** | **54.44** | **60.8** | 6.2 | **29.6** | 0.405 | **34.25** |
| BLIP-2 T5 | base | 29.71 | 33.37 | 43.8 | **2.4** | 15.6 | **0.575** | 10.12 |
| | ours | **50.91** | **45.24** | **60.8** | 7.3 | **26.3** | 0.389 | **43.56** |
| PaliGemma | base | 26.63 | 31.02 | 54.9 | **3.8** | 17.7 | **0.616** | 10.29 |
| | ours | **58.57** | **50.91** | **57.9** | 4.3 | **25.1** | 0.482 | **23.41** |
| LLaVA | | 74.41 | 61.23 | 62.8 | 6.3 | 36.7 | 0.429 | 53.48 |

Table 3: Left: Recall@1 results on text-to-image and image-to-text retrieval. Right: Hallucination results on the COCO validation set, and DCI benchmarks.

VLMs to longer, high-probability sequences. Still, our method of EOS token debiasing proves considerably more effective than this inference-time approach, outperforming it across nearly all models and datasets. The sole exception is BLIP-2 OPT on FineCapEval, where the two approaches perform on par. The performance gain is most pronounced for BLIP-2 T5, showing a significant boost across all datasets.

The coherence metric shows a nuanced pattern, indicating a possible interaction between caption length and coherence. While for most model-dataset combinations, coherence remains stable or even improves with EOS debiasing, for BLIP-2 OPT, we see a dip in coherence on images from DCI and DOCCI. We discuss the outlier behavior of BLIP-2 OPT in more detail in §5.2. This metric also uncovers a major issue with the trivial solution: coherence suffers greatly across all models and datasets. The qualitative analysis in §4.5 reveals that the EOS blocking actually results in an invalid solution to the task of detailed image captioning. We attribute the superior performance of EOS debiasing over EOS blocking to the gradual shift in distribution, and the greater freedom of exploration which sequence training enables.

Looking at the state-of-the-art TinyLLaVA scores, we see that simple EOS debiasing is able to close much of the performance gap to this instruction-tuned model trained on massive data. This result supports our hypothesis that much of the capabilities needed for detailed image captioning are already present in the vision and language backbones of the base models, but also indicates that there is a limit which can only be overcome through extensive supervised finetuning.

**Retrieval** results on the Urban-1k dataset are reported in Table 3 (left side). We observe impressive improvement over the baseline for both text-to-image and image-to-text retrieval. The text-to-image recall for PaliGemma, for example, more than doubles after EOS token debiasing. The extra details induced appear to be highly relevant to the input image. They enable the retriever to better identify the target image from among many visually similar urban scenes, in line with the discriminability objective outlined in Luo et al. (2018).

**Hallucination** results are reported in Table 3 (right.) They indicate that the increase in descriptiveness comes with an expected increase in the rate of hallucination (Zhou et al., 2024b). This tendency is likely to be inflated in the CHAIR$_i$ scores as any object mentioned in the generated caption that is not in the short COCO reference is deemed a hallucination. The CHAIR$_i$ scores can thus be interpreted only in reference to the relative recall achieved by a model, as a ratio of hallucinated objects to correctly mentioned objects. We see that on these metrics, the two BLIP-2 VLMs yields largely comparable scores to TinyLlaVA.

Shifting focus to the ALOHa scores measured on DCI, we see that our three models and TinyLLaVA all score in a similar range for both recall and hallucination. From these results, we can conclude that EOS debiasing induces a similar hallucinatory behavior to explicit finetuning for the task of detailed image captioning. Even in the absence of explicit guidance towards relevant detail in EOS debiasing, the models find solutions consistent with their visually-grounded pretraining.

| Image | Model | | Captions |
|---|---|---|---|
| | Reference | | Two kids one boy and one girl standing in a lawn surrounded by trees and watching behind with house and lighting visible at the top |
| | BLIP-2 OPT | base | a couple of children are hugging each other in front of a christmas tree |
| | | triv. | a couple of kids hugging each other in front of a christmas tree\n\na boy and a girl are hugging in the middle of the street\n\n\n\t\t[IMAGE]\n\t [IMG]http |
| | | ours | a little girl and boy and girl are hugging each other as they look at a christmas tree in front of a house with lights |
| | BLIP-2 T5 | base | a little boy and girl hugging in front of some trees |
| | | triv. | a boy and girl hugging each other in front of bushes christmas trees santa claus the christmas tree two boys huggin' eachother in the front yard |
| | | ours | a little boy and girl are standing in front of some christmas trees on the side of the road. they are hugging each other as they look up at the tree tops. there are some houses in the background that are lit up with christmas lights |
| | PaliGemma | base | Two young children hugging each other in front of a Christmas tree |
| | | triv. | A couple of kids standing next to each other. A boy and a girl hugging each other's shoulders. \n THERE IS A PICTURE OF A BOY AND A GIRL HUGGING EACH OTHER |
| | | ours | Two children hugging each other while standing next to a Christmas tree in front of a house with lights on it |

Table 4: Captions generated by base models (base), with trivial EOS blocking (triv.) and with EOS debiasing (ours). The image is from the FineCapEval dataset.

## 4.5 Qualitative Results

Table 4 presents a comparison between captions generated by our models (highlighted in orange) and their respective base versions. Our models consistently enrich the captions with additional meaningful details. The captions not only mention the presence of a boy and a girl but also incorporate contextual elements such as house lights and Christmas decorations. Interestingly, these enhancements are not merely appended at the end but are integrated naturally throughout the captions. An illustration of how captions evolve over training steps can be found in Appendix A.2.

Although the trivial solution showed high performance in terms of CAPTURE scores in Table 2, here we see that it results in incoherent, repetitive captions. In fact, the generated text appears to consist of several short generic captions strung together.

# 5 Analysis and Discussion

In this section we present training details, discuss differences in model behavior under EOS debiasing, and discuss the mechanism behind EOS debiasing.

## 5.1 Training Progression

Given the unusual nature of our training objective, here we present details on the training progression of EOS debiasing, using BLIP-2 T5 as an example. Figure 1a illustrates training progression in terms of caption length and unigram recall over a random sample of 500 data points from the COCO validation set. We see that right from the start, increasing length yields higher recall, i.e. the extra detail added to the generated caption is relevant to the input image. The length progression starts off slow, then about 75% through the epoch spikes, with recall closely following.

Figure 1b shows how the EOS probability and the rest-of-sequence (ROS) probability change during training. We see that while the ROS probability is mostly stable, the EOS token probability drops initially, as expected, then increases substantially and subsequently decreases again (this point coincides with the rapid increase in recall and length observed

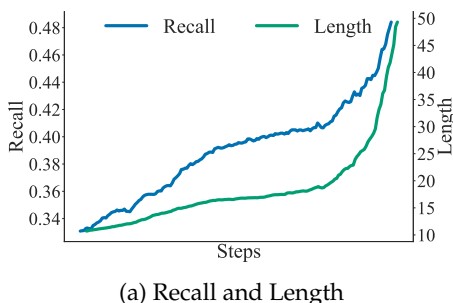

(a) Recall and Length

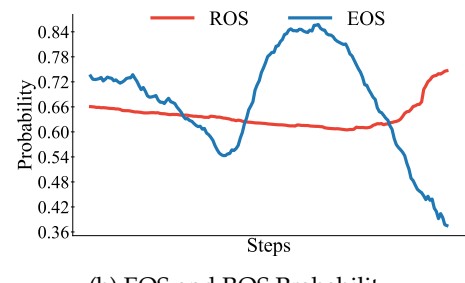

(b) EOS and ROS Probability

Figure 1: Training progression of BLIP-2 T5 as measured on COCO validation set.

| Model | DCI | | |
|---|---|---|---|
| | CIDEr ↑ | Recall ↑ | CAPTURE ↑ |
| Base | 1.21 | 15.6 | 34.62 |
| CP1 | 4.37 | 20.14 | 42.23 |
| CP2 | 5.63 | 21.28 | 43.07 |
| CP3 | 6.34 | 21.93 | 43.77 |
| CP4 | 8.72 | 26.31 | 46.47 |

Table 5: DCI results for different BLIP-2 T5 checkpoints.

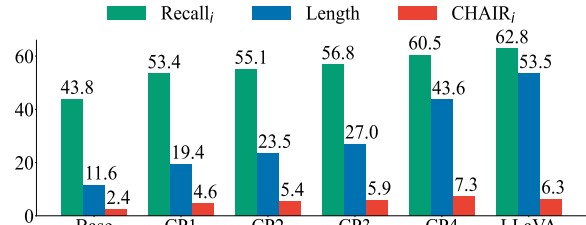

Figure 2: Comparison across different checkpoints against TinyLLaVA on COCO validation set.

in Figure 1a). This wave-like pattern cannot be trivially explained and likely has to do with the complex dynamics of sequence training.

Figure 2 compares four checkpoints from the later training stages (steps 6000, 7500, 8400, and 9150), the base BLIP-2 T5 model, and TinyLLaVA in terms of Recall, $CHAIR_i$ hallucination, and sequence length. From the base model to the first checkpoint, the increase in Recall and caption length is significant. This improvement trend continues through subsequent checkpoints, accompanied by an inevitable increase in hallucinations as the sequence length grows. However, using TinyLLaVA as a reference, the increase in $CHAIR_i$ hallucination remains moderate, especially as compared to the substantial gains in descriptiveness. Table 5 further shows the performance of the four checkpoints on the DCI dataset. The boost in CAPTURE and Recall persists across all checkpoints, with CP4 showing significant gain over earlier checkpoints.

## 5.2 Model Behavior

In Table 2 we see that the three models exhibit different behavior when undergoing EOS debiasing. BLIP-2 T5 has the lowest base performance in terms of CAPTURE to begin with, but after EOS debiasing, it ends up leading on DCI and DOCCI. The improvement is echoed in the CIDEr and Coherence scores. This substantial performance boost can be attributed to its encoder-decoder architecture, where the bidirectional attention mechanism in the encoder helps penalize fine-grained patterns that lead to premature termination.

BLIP-2 OPT also shows an increase in performance as a result of EOS debiasing, albeit with a notable decline in Coherence. Upon closer inspection, we identified a mismatch between the EOS token used in the VLM pre-training and the one originally defined in the OPT vocabulary. The EOS token used in BLIP-2 OPT has ID 50118, corresponding to the newline "\n" token, while for OPT, the EOS token has ID 2 which corresponds to token "". This discrepancy likely contributes to the drop in Coherence, as the longer debiased sequences are not effectively regulated by the linguistic capabilities of the language modeling backbone.

PaliGemma shows the smallest increase in caption length among the three models, albeit with stable Coherence scores and good gains in CAPTURE scores. This indicates a strong grounding capability, likely due to its end to end training. This is further reflected in the retrieval results in Table 3 where we see that EOS-debiased PaliGemma captions yield the best discriminability in text-image matching.

### 5.3 The Mechanism behind EOS Debiasing

One could wonder why EOS debiasing does not result in a trivial and undesired solution where extra detail is added only at the end of the sequence. Since we are using sequence training rather than cross-entropy over a single token, penalizing the EOS token does not simply discourage the occurrence of this one token but suppresses an entire subspace of sequences that are likely to terminate early. This suppression fundamentally alters the token-level probability landscape, reshaping the model's learned sequence distributions. A key factor in this behavior is the self-attention mechanism, which dynamically modulates token interactions during generation. Each token's attention distribution varies across training samples, leading to shifts in how the model contextualizes EOS token predictions. Specifically, when an EOS penalty is applied, the model learns to redistribute probability mass toward sequences that are structurally different from those leading to early termination. As a result, the model does not trivially postpone EOS but instead restructures its output, incorporating additional details within the sequence rather than appending them superficially at the end.

## 6 Conclusion

This work introduced EOS token debiasing, an unsupervised sequence training method for detailed image captioning which can be applied to any pre-trained VLM. By penalizing early sequence termination, EOS debiasing encourages the model to generate longer, more relevant captions with richer detail. Our results demonstrate that this approach strikes a balance between enhanced descriptiveness, coherence, and correctness, leading to more informative and contextually accurate captions. The analysis of training progression and EOS probability dynamics revealed important insights into model behavior, emphasizing the role of sequence structure in generating meaningful outputs. Overall, EOS debiasing reshapes the model's output distribution, fostering more natural and detailed captioning. The method can be adapted to any task where the available training data is short and generic, and a base model is available that has some internal knowledge about the task from pretraining, that would be suppressed during standard finetuning on short data. We invite future work to test the efficacy of EOS debiasing in diverse conditional generation tasks, as a complement or predecessor to supervised finetuning.

## Limitations

A key limitation of EOS debiasing is that while it induces longer and more detailed captions from a pretrained VLM without any explicit training for the task, there appears to be a limit to the enrichment that can be achieved. We see that prolonged training results in an increase in hallucinations and repetitions. The method is therefore suitable for settings where no finetuning data is available for detailed image captioning. When such data are present at hand, we recommend performing brief EOS debiasing followed by supervised finetuning, thus first unlocking the detailed captioning capabilities of the base VLM for better utilization of the finetuning data.

## Acknowledgments

We thank Jameel Hassan for his contribution in the early stages of this work.

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

# A  Appendix

## A.1  Extra Details

Table 6 shows an image from the COCO training set. As shown, the majority of the added details did not appear in the training captions, which hints at the ability of the model to extrapolate beyond information seen in the training data, adding new details relevant to the input image.

| Image | Model | Captions |
|---|---|---|
|  | References | A man in orange jersey throwing a baseball. \| A young baseball player is on the field with a mitt. \| A baseball pitcher steps forward as he winds up a pitch. \| A pitcher is winding up for a pitch. \| A pitcher in mid pitch at a baseball game. |
| | Blip-2 OPT | a young man in an orange baseball uniform throwing a baseball |
| | | a young man in an image of a young baseball player pitching a ball on the mound throwing a pitch during a game at a baseball game with a scoreboard in the background |
| | Blip-2 T5 | a baseball player is pitching the ball on the mound |
| | | a young baseball player in an orange and white baseball uniform throws the ball from the pitcher's mound to the batter on the other side of the field. he is about to deliver the pitch when the yellow jackets scoreboard appears behind him |
| | PaliGemma | A baseball player pitching a ball on top of a field. |
| | | A baseball player on the mound about to ready to throw the ball during a baseball game with the scoreboard in the background |

Table 6: Captions generated by models before and after (orange) EOS debiasing on an example from the COCO training set. Many of the extra details added are not in the reference captions.

## A.2  Progression of Captions

The progression of the details is quite interesting. As shown in Table 7, the extra details are added in different parts of the caption. Moreover, we see that length grows in a non-linear fashion across equally spaced training steps.

| Image | Model | Captions |
|---|---|---|
| | | a dining room table and chairs are in the middle of the room |
| | | a dining room table and chairs are in the middle of the living room |
| | | a dining room table is in the middle of the room next to the kitchen and living room |
| | | a dining room table is set in the corner of the room next to the kitchen and living room |
| | Blip-T5 | a dining room table is in the middle of the room |
| | | a dining room table is set in the corner of the living room and kitchen. A blue tablecloth is on the table. There is an open doorway to the dining area from the kitchen, which has white cabinets and stainless steel appliances |
| | | a dining room table is set in the corner of the living room and kitchen. A green tablecloth covers the table and chairs are arranged around it. There is an open doorway leading into the kitchen from the dining area |
| | | a dining room table sits in the corner of the room next to the kitchen cabinetry. A green tablecloth is on the table and chairs are arranged around it. There is an open doorway leading into the living room from the dining area |
| | | a dining room table sits in the middle of the living room. A blue and green tablecloth is on the table. There is an open doorway into the kitchen from the dining area. The refrigerator, dishwasher, oven, microwave, and sink are all in plain view |

Table 7: Progression of generated caption when training with EOS debiasing.

## A.3 Ablation

### A.3.1 Trivial baseline

For completeness, Table 8 compares the trivial solution under different decoding strategies: beam search, contrastive decoding (Su & Collier, 2022), and nucleus sampling (Holtzman et al., 2019), against our approach. The beam size for beam search is set to 5, the contrastive decoding parameters are $\alpha = 0.7$ and $k = 5$, and the top-$p$ for nucleus sampling is 0.9.

| Model | FineCapEval | | DCI | | DOCCI | |
|---|---|---|---|---|---|---|
| | CAPT ↑ | Coh ↑ | CAPT ↑ | Coh ↑ | CAPT ↑ | Coh ↑ |
| beam search triv. | **46.72** | 2.05 | 43.51 | 2.14 | 42.13 | 2.15 |
| contrastive triv. | 44.36 | 2.13 | 45.76 | 2.31 | 45.78 | 2.21 |
| nucleus triv. | 43.34 | 1.97 | 44.07 | 2.16 | 44.40 | 2.18 |
| ours | 46.52 | **2.87** | **46.47** | **2.41** | **47.13** | **2.54** |

Table 8: Comparison between different trivial baselines' CAPTURE and coherence scores on BLIP-2 OPT.

### A.3.2 Learning Rate

To demonstrate the stability of our approach, Table 9 shows the performance of BLIP-2 T5 using different learning rates. As shown, performance gains are consistent across learning rates, with 1e−7 yielding the best results while also ensuring fast convergence.

| Learning rate | FineCapEval | DCI | DOCCI |
|---|---|---|---|
| 5e−7 | 46.71 | **46.90** | 45.80 |
| 1e−7 | 46.52 | 46.47 | **47.13** |
| 5e−8 | **47.82** | 45.34 | 44.62 |

Table 9: CAPTURE scores for BLIP-2 T5 variants across different learning rates.

### A.4 GPT-4o Prompt

Table 10 shows the prompt used to induce Coherence scores from GPT-4o. We use a temperature of 1.0 when calling the API.

> Your task is to rate the coherence of one image caption presented between quotation marks. Output an evaluation score on a scale from 1 to 5 based on the following criteria and rating scale:
> Coherence: The caption should be well-structured and well-organized. It should build from phrase to phrase or from sentence to sentence to a coherent body of information about the image.
> 1 Incoherent: The caption lacks structure and logical flow; sentences/phrases are disjointed or unrelated.
> 2 Weakly coherent: Some connections between phrases/sentences exist, but the overall organization is unclear or inconsistent.
> 3 Moderately coherent: The caption conveys information with a basic level of organization, though some parts may be unclear or loosely connected.
> 4 Mostly Coherent: The caption is well-structured and flows logically, with minor inconsistencies or awkward transitions.
> 5 Fully coherent: The caption is highly structured, well-organized, and presents a clear, logical progression of information.
> Caption: {generated}.

Table 10: Prompt used for evaluating captions coherence with GPT-4o.

