# OpenReview forum: "The Devil is in the EOS: Sequence Training for Detailed Image Captioning"
_colmweb.org/COLM/2025/Conference — COLM 2025_

### Official Review · Reviewer_m9TX · 2025-05-13

**Rating:** 6
**Confidence:** 4
**Ethics Flag:** 1

**Summary:**

The authors propose to elicit more detailed image captions by carefully minimizing the probability of the end-of-sequence (EOS) token via finetuning on generated captions.

The approach generally improves CAPTURE-based word f1 estimates on detailed captioning tasks relative to model baselines trained on simple captions (MSCOCO), outperforming both baseline performance, and EOS blocking (base and triv, Table 2)

However coherence scores (as determined by GPT-4) degrade significantly on 2 out of 3 tasks for all models relative to baselines, although much less than EOS blocking (triv, Table 2).

Results also indicate that, as expected, longer more detailed captions both boost retrieval recall, and increase hallucination rates to some extent (Table 3).

Generally results are significantly worse than a SOTA model trained on detailed image captions, but significantly better than baselines that are also trained only on simple captions (Llava model, Table 2 and 3). Qualitative results are encouraging (table 4).

While results as a function of checkpoint performance indicate steady improvement (Table 5, Figure 2), commentary in the appendix indicates that a very specific learning rate (1e-7) was required to achieve good results.

**Questions To Authors:**

See previous section.

**Reasons To Accept:**

- The approach is simple and is shown to be effective at improving detailed captioning performance in the scenario that only simple captioning training data is available.
- The experiments consider multiple models, detailed captioning datasets, and performance metrics in their study.

**Reasons To Reject:**

- As mentioned, the appendix indicates that very specific learning rates were required for success. While some investigations into how performance evolves as a function of training iteration were conducted (Figure 1), more discussion and experimentation is warranted. With early-stopping based on key metrics, I'd expect that the approach would be stable over at least a limited range of learning rates (and without, always degrade the model eventually). Authors, please comment. This is important in establishing the utility of the approach.

- When detailed captioning data is available, the authors propose to "warm-start" models with their EOS debiasing method before finetuning on the detailed captioning data, but no experiments are presented. Considering that high-quality detailed captions (annotated and synthetic) are quite readily available now, this seems like an important result to establish,
 but this is outside the scope of the current paper.

- There are quite a few small mistakes throughout the paper. Many are typos, but some are important. For example equation (2), which defines their method, is poorly defined, and has the wrong sign. Likely the "i" subscript be "n" to denote the last generated token (but penalizing EOS for all tokens would also work). What did the authors do?

---

> ### Author Response · Authors · 2025-06-03
>
> We thank Reviewer m9TX for their thoughtful comments and careful reading of our submission.
>
> > 1- As mentioned, the appendix indicates that very specific learning rates were required for success. While some investigations into how performance evolves as ...
>
> The eventual degradation of the model is observed across all learning rates we explored and is a consequence of the progressive suppression of the EOS token, which in the limit is bound to lead to undesirable behavior (i.e. when there is no additional relevant detail that the model can add, it can only satisfy the training objective through repetition and hallucination.)  As such, the early stopping mechanism proves more crucial to the success of the approach than the choice of learning rate itself. The learning rate of 1e-7 is the result of empirical hyperparameter tuning, similar to how that would be done in any standard finetuning framework (where one learning rate would work better than another).
>
> The commentary in the appendix is merely a speculation on why the learning rate we adopted (1e-7) proved optimal. Our observation is that with higher learning rates, degradation occurs sooner thus leaving the model less room to search for an optimal solution, and lower learning rates produce very similar final performance to the optimal learning rate, but require significantly more training steps to reach comparable quality. That being said, both higher and lower learning rates result in more detailed captions being generated at some stage of the training process, which can be identified by the early stopping mechanism. We report results with two alternative learning rates: 5e-7 and 5e-8.
> The results for 5e-7 compared to 1e-7 indicate minor fluctuations in CAPTURE score with 1e-7 proving a bit better:
>
> | Model           	| FineCapEval CAPT ↑ | DCI CAPT ↑ | DOCCI CAPT ↑ |
> |---------------------|-------------------|-------------|----------------|
> | BLIP-2 T5 ($1\mathrm{e}{-7}$)   | 46.52         	| 46.47   	| 47.13      	|
> | BLIP-2 T5 ($5\mathrm{e}{-7}$)  | 46.71        	| 46.90   	|45.8       	|
> | BLIP-2 T5 ($5\mathrm{e}{-8}$)  | 47.82        	| 45.34   	|44.62       	|*
>
>
>
> > 2-When detailed captioning data is available, the authors propose to "warm-start" models with their EOS debiasing method before finetuning ...
>
> Indeed, we judged this experiment to be outside the scope of _this_ paper, because we are particularly interested in exploring this proposal in the context of multilingual detailed image captioning, where high-quality detailed captioning data may not be as abundant as it is in English. In that context in particular, we expect the “warm-start” to prove highly effective.
> At the moment, detailed image captioning data for non-English languages is virtually non-existent to the best of our knowledge. However, we are aware of (and involved with) efforts to close this resource gap, so upon completion of these ongoing data collection projects, we plan to test the potential of our EOS debiasing method for warm-starting model finetuning on multilingual detailed image captioning data.
>
> > 3-There are quite a few small mistakes throughout the paper. Many are typos, but some are important. For example equation (2), which defines their method, is poorly defined, and has the wrong sign. Likely the "i" subscript be "n" to denote the last generated token (but penalizing EOS for all tokens would also work). What did the authors do?
>
> Thank you for catching this issue. The subscript in Equation (2) should explicitly denote the final token index (n), not i, and the sign should be positive, as we are penalizing the log-probability of the EOS token.
> \begin{equation}
>  	\nabla \mathcal{L}(\theta) = \nabla_\theta \log  p_\theta(t_n)
> \end{equation}
> where $t_n$ denotes the EOS token.
>
> An alternative formulation we considered is the following (The equation uses cases which is not supported in openreview):
>
> $$
> \nabla L(\theta) =
> \begin{cases}
> \nabla_\theta \log p_\theta(t_i) & \text{if } t_i = \text{EOS} \\ \quad
> 0 & \text{otherwise}
> \end{cases}
> $$
> We welcome feedback from the reviewer on whether this formulation is clearer and should instead be used in the final draft.
> The intention behind this formulation is to communicate that the penalty only applies to the EOS token itself, while the gradients for all the other tokens are masked out. The non-EOS tokens are implicitly affected through the self-attention mechanism, but there is no explicit penalty applied to them.
> We will closely review the draft and correct any typos.

---

> > ### Comment · Reviewer_881E · 2025-06-08
> > **Confusion on eq 2 explanation**
> >
> > Hi authors, you provided a fixed equation two and an alternative formulation.
> > Can I ask what is the difference between the fixed one and alternative one. They look the same to me. thanks!

---

> > > ### Author Response · Authors · 2025-06-08
> > >
> > > Dear reviewer, the two equations are indeed equivalent. We presented both version just to take your advice on which one seems clearer.
> > >
> > > Please let us know if our response successfully addresses your concerns and thank you for your time.

---

> ### Comment · Reviewer_m9TX · 2025-06-09
> **Significance and utility of the technique in the unsupervised setting still needs to be established**
>
> Thank you for the response authors.
>
> However, with warm-start out of scope, the significance and utility of the technique in the unsupervised setting still needs to be established.
>
> That means establishing that the metrics used for early stopping on coarse data can be reliably utilized to improve target metrics on detailed captioning data.
>
> Currently the paper does not seem to contain any results or analysis that directly demonstrates this.
>
> Transparent results showing how key metrics evolve on both coarse and detailed holdout data as a function of EOS-debiasing finetuning iteration will hopefully reveal that this is possible, and inform a procedure for how the technique can be utilized in practice.

---

> > ### Author Response · Authors · 2025-06-09
> >
> > Dear reviewer, we chose to perform model selection using coarse data precisely because in an unsupervised setting (which is the main setting our method targets) detailed heldout data for validation would not be available. We can report results on the evolution of key metrics on coarse and detailed held out data, but the motivation behind this request is not clear to us. Our model selection process, based on coarse validation data, results in a checkpoint with strong results. Even if a stronger checkpoint could be selected based on detailed validation data, (1) does this invalidate our strong results and (2) does this not violate the unsupervised training assumption?

---

> > > ### Comment · Reviewer_m9TX · 2025-06-09
> > >
> > > Thanks, it is good to know that model selection is done based on coarse data. Is this stated in the paper? More generally, is your model selection process described in the paper? If so that's great, please point me to it. If not, providing this is clearly the number one priority.
> > >
> > > My request is meant to 1) elicit full transparency wrt your model selection process (what data, what metrics), the stability of the approach, and how downstream performance evolves as EOS debiasing is done. These are central questions, are they not?

---

> > > > ### Author Response · Authors · 2025-06-10
> > > >
> > > > Thank you for the response and for spotting a critical omission in our write up. The relevant results are actually in the paper but the discussion on model selection is indeed missing. We discuss this matter below and will include the discussion in the final paper too.
> > > >
> > > > Figure 2 and Table 5 show how key metrics evolve on both coarse and detailed heldout data as a function of EOS-debiasing finetuning. We repeat the relevant numbers in the table below:
> > > >
> > > > | Checkpoint           	| COCO Val. Recall_i ↑ | DCI CAPTURE ↑ |
> > > > |---------------------|-------------------|-------------|
> > > > |  Base  | 43.8 | 34.6   	|
> > > > | CP1  |  53.4 | 42.2 |
> > > > | CP2 | 55.1 | 43.1 |
> > > > | CP3 | 56.8 | 43.8 |
> > > > | CP4 | 60.5 | 46. 5|*
> > > >
> > > > Recall_i is a computationally cheap metrics, which makes it suitable for iterative validation during training.
> > > >
> > > > We see that the same upward trend holds between the coarse-caption scores and the detailed-caption scores, with even the magnitude of change matching between the two. While we have not been able to train models until saturation in the validation recall score (due to computational constraints on the maximum output length we can fit on the GPU card we're working with), we expect that saturation on the course validation data would correspond to saturation on the detailed data as well. The reasoning here is that while COCO captions contain less detail on a per-caption basis, on a global scale they still cover a lot of different objects, attributes and relations, so it provides good coverage for the details that are likely to be mentioned in a fine-grained caption.

---

> > > > > ### Comment · Reviewer_m9TX · 2025-06-10
> > > > >
> > > > > Thank you for the response. So effectively, early stopping has not been investigated. It's not clear that coarse recall would saturate before detailed metrics degraded significantly. Anyway, I appreciate that early stopping is challenging, considering the mismatch between coarse and detailed captioning data. Are coherence scores on detailed data perhaps indicative of when to stop? A small subset of detailed data could be used to assess the checkpoints...

---

> > > > > > ### Author Response · Authors · 2025-06-10
> > > > > >
> > > > > > Thanks for following up. Coherence scores would unfortunately not work -- as discussed in our response to reviewer Pbv2 (weakness #3), differences in coherence scores are meaningful when measured on captions of comparable length (i.e. in the comparison of detailed captioning models), but across different caption lengths, coherence scores tend to favor shorter captions (i.e. the metric would not be suitable for model selection during training, where caption length progressively increases.)
> > > > > >
> > > > > > That being said, if one is willing to assume the availablity of some detailed validation data, early stopping can easily be done based on CAPTURE scores. We long debated whether this would be a valid approach or a violation of the unsupervised experimental setup. From the discussion above, we conclude that this may indeed be the best approach to take here. We will include a further exploration of this point in the final version of the paper.
> > > > > >
> > > > > > With the discussion period reaching its end, we want to thank you for the insightful feedback and positive engagement. If our responses have sufficiently addressed your concerns to consider the paper above the acceptance threshold, please consider increasing your score. We remain committed to implementing all changes discussed above.

---

> > > > > > > ### Comment · Reviewer_m9TX · 2025-06-10
> > > > > > >
> > > > > > > Thank you authors. While the scope of the paper is somewhat limited, the idea is interesting, and the paper is sound. I'll increase my score from 5 to 6.

---

### Official Review · Reviewer_881E · 2025-05-13

**Rating:** 7
**Confidence:** 4
**Ethics Flag:** 1

**Summary:**

In this paper, the author proposes a data-free sequence-training method EOS debiasing to finetune VLMs to improve their performance on detailed caption generation. The authors validate their method on three VLMs and three benchmarks. Compared to strong baseline, the authors show increase on caption length and details wile maintaining moderate increase on hallucinations.

**Questions To Authors:**

Is there a way to interpret the loss as reward and reinforcement learning?

**Reasons To Accept:**

- The method is very simple and intuitive. It should be easy to reimplement and can be plug and play on any language models including decoder only and encoder-decoder models.
- The candidate did extensive study on the method. They validate their method on multiple models and multiple datasets.
- The candidate also provide many analysis on lenght/recall/EOSROS probability to show what change the proposed method would bring to the model.

**Reasons To Reject:**

- It is not clear to me how this methods can help other tasks.
- The method itself is useful when there is no fine-tuning data but the benefit may be less when there are fine-tuning data.
- Based on the “mechanism behind EOS debiasing”, there should be other sequence training method that potentially work: for example, use length as reward. The authors should explore more variants.

---

> ### Author Response · Authors · 2025-06-02
>
> Dear reviewer 881E,
>
> We thank the reviewer for the detailed and thoughtful feedback. Below, we address each point raised.
>
> > It is not clear to me how this methods can help other tasks.
>
> The method can be adapted to any task where the available training data is short and generic, and where the model has some internal knowledge about the task from pretraining. It helps to utilize the model's latent knowledge that may be suppressed during finetuning. This should be applicable not only to the space of vision-language modeling but in the context of conditional generation in general. We will make this point more explicit in the conclusion of the paper, to encourage further investigation into the potential of the method.
>
> > Is there a way to interpret the loss as reward and reinforcement learning?
>
> While our approach does not explicitly follow a reinforcement learning framework, there is a conceptual connection in the sense that we are modifying the model’s behavior based on a form of task-specific signal, i.e. we are discouraging premature termination by penalizing the EOS token logits. Unlike RL where rewards would be dynamically generated by an external function, however, we perform direct gradient-based optimization on the logits with a deterministic penalty, based on the learning rate.
> > The method itself is useful when there is no fine-tuning data but the benefit may be less when there are fine-tuning data.
>
> This is a valid point, our method is primarily designed for low-resource settings where only generic data (e.g., MS-COCO) is available. Yet, as mentioned in the paper, the method can also serve as a warm start for finetuning on detailed image captioning data. We plan to explore this direction in future work, as discussed further in our response to reviewer m9TX.
>
> > Based on the “mechanism behind EOS debiasing”, there should be other sequence training method that potentially work: for example, use length as reward. The authors should explore more variants.
>
> We agree that alternative strategies, such as optimizing for caption length via reinforcement learning, may be viable. Our proposed approach is informed by prior findings on the role of the EOS token in controlling the length of generated text and, as such, it is a more direct intervention than rewarding length would be. Furthermore, it avoids the need for a baseline decoding strategy (e.g., greedy decoding) to reduce variance in the reward, which would be required when optimizing for length due to its variability.
>
> > Is there a way to interpret the loss as reward and reinforcement learning?
>
> While our approach does not explicitly follow a reinforcement learning framework, there is a conceptual connection in the sense that we are modifying the model’s behavior based on a form of task-specific signal, i.e. we are discouraging premature termination by penalizing the EOS token logits. Unlike RL where rewards would be dynamically generated by an external function, however, we perform direct gradient-based optimization on the logits with a deterministic penalty, based on the learning rate.

---

> > ### Comment · Reviewer_881E · 2025-06-10
> >
> > The comments well addressed my questions. I remain my original rating.

---

### Official Review · Reviewer_Pbv2 · 2025-05-13

**Rating:** 7
**Confidence:** 4
**Ethics Flag:** 1

**Summary:**

The manuscript introduces EOS debiasing, an unsupervised sequence-training method that gradually penalises the probability of the end-of-sequence (eos) token in pretrained vision-language models (VLMs). The authors argue that generic caption corpora bias models toward prematurely outputting <eos>, yielding short, coarse captions. By fine-tuning only the cross-modal bridge on MS-COCO and applying the debiasing objective, they obtain longer captions on three fine-grained caption benchmarks (FineCapEval, DCI, DOCCI) and report improvements in CAPTURE, CIDEr (for shorter captions), retrieval recall, and human-judged coherence, while acknowledging a rise in hallucination metrics. Experiments cover three mid-sized VLMs and compare to a trivial EOS-blocking at inference baseline and to TinyLLaVA as SOTA. An analysis section explores training dynamics and the trade-off between length, recall and hallucination.


--------------------After rebuttal--------------------

Many thanks for the rebuttal, which further addresses my concerns. I decided to keep my score.

**Questions To Authors:**

Please see above for details. Thanks.

**Reasons To Accept:**

1. The motivation is clear.
2. The paper is well-written.
3. The experiments are extensive.
4. The method achieves improved performance on multiple baselines and datasets.
5. The analysis is insightful.

**Reasons To Reject:**

1. The baselines used for comparison are limited. Only “EOS-blocking” and Tiny-LLaVA are used. No comparison to several important baselines, such as self-critical RL with CIDEr reward, CLIP-reward, or SMILE.


2. The “EOS blocking” baseline still uses beam search with default length penalties; alternative decoding strategies (top-p + length penalty, contrastive search with length-bias) could yield stronger baselines and reduce coherence collapse

3. The GPT-4o coherence score appears unreliable. The evaluation prompt is provided in the appendix, and the paper reports neither inter-annotator agreement nor the sampling temperature. An LLM-as-judge often rewards verbosity, so the metric may be biased toward the proposed method’s longer captions.

---

> ### Author Response · Authors · 2025-06-03
>
> Dear reviewer Pbv2,
>
> We thank the reviewer for the helpful comments and suggestions. Below, we provide our responses to each point.
>
> > 1-  The baselines used for comparison are limited. Only “EOS-blocking” and Tiny-LLaVA are used. No comparison to several important baselines, such as self-critical RL with CIDEr reward, CLIP-reward, or SMILE.
>
> We agree that comparing against additional baselines would help position our method more clearly to previous work. However, we previously faced issues in running the codebases for SMILE and CLIP-Reward and were not able to replicate the expected results for the latter through a reimplementation. We are exploring this further at the moment and will share our progress below. Regarding self-critical sequence training with CIDEr reward, we note that it aims to improve CIDEr scores on MS-COCO, but is not designed to yield more descriptive captions, which makes it less suitable for our focus on fine-grained captioning.
>
> > 2- The “EOS blocking” baseline still uses beam search with default length penalties; alternative decoding strategies (top-p + length penalty, contrastive search with length-bias) could yield stronger baselines and reduce coherence collapse
>
> Thank you for the suggestion. Our intention was to have the same settings for EOS debiasing and the EOS blocking. We experimented with adding a length penalty (see table below). Contrastive decoding does yield better performance than EOS blocking with beam search in most cases (except for the CAPT score on FineCapEval). However, coherence remains low, with similar artifacts and caption stitching patterns as observed in earlier experiments. We will include these additional results in the appendix of the final draft.
>
> | Model                            | FineCapEval CAPT ↑ | FineCapEval Coh ↑ | DCI CAPT ↑ | DCI Coh ↑ | DOCCI CAPT ↑ | DOCCI Coh ↑ |
> |----------------------------------|---------------------|--------------------|-------------|------------|----------------|---------------|
> | BLIP-2 OPT (triv.)               | 46.72               | 2.05               | 43.51       | 2.14      | 42.13          | 2.15          |
> | BLIP-2 OPT (contrastive triv.)   | 44.36               | 2.13               | 45.76       | 2.31       | 45.78          | 2.21          |
> | BLIP-2 OPT (nucleus triv.)       | 43.34               | 1.97               | 44.07       | 2.16       | 44.40          | 2.18          |
> | BLIP-2 OPT (ours)                | 46.52               | 2.87               | 46.47       | 2.41       | 47.13          | 2.54          |*
> .
> > 3- The GPT-4o coherence score appears unreliable. The evaluation prompt is provided in the appendix, and the paper reports neither inter-annotator agreement nor the sampling temperature. An LLM-as-judge often rewards verbosity, so the metric may be biased toward the proposed method’s longer captions.
>
> The reviewer expresses concern that the metric may reward verbosity, but the numbers reported in Table 2 do not support such a view: in 6 out of 9 cases, coherence scores favor the _base_ captions, which are considerably shorter than the captions obtained with the trivial baseline or our method. In our view, this is not surprising as shorter text offers fewer opportunities for error and typically exhibits a simpler discourse structure.
>
> Between the lengthier captions generated by the trivial baseline and our method, we observe lower coherence scores for the former, which can be easily explained with reference to the examples shown in Table 4: the trivial baseline achieves higher caption length through a hack, wherein in generates 2 or 3 separate short captions, stringed together. Naturally, this results in lower coherence, since there is unnecessary repetition and a lack of connectives between the sentences.
>
> Regarding the sampling temperature, all methods were evaluated under the same temperature of 1.0. We will include this in the final draft.

---

> > ### Comment · Reviewer_Pbv2 · 2025-06-10
> >
> > Thanks for the rebuttal, which further addresses my concerns. I decided to keep my score.

---

### Comment · Program_Chairs · 2025-04-03

This paper violates the page limit due to adding a limitation sections beyond the page limit. COLM does not have a special provision to allow for an additional page for the limitations section. However, due to this misunderstanding being widespread, the PCs decided to show leniency this year only. Reviewers and ACs are asked to ignore any limitation section content that is beyond the 9 page limit. Authors cannot refer reviewers to this content during the discussion period, and they are not to expect this content to be read.

---

### Decision · Program_Chairs · 2025-07-08

**Decision:**

Accept

**Comment:**

Meta Review of **The Devil is in the EOS: Sequence Training for Detailed Image Captioning**

This submission highlights that base VLMs often generate short, undescriptive and generic captions when evaluated on image captioning. The submission proposes the argument that this is due to a strong bias to generate the EOS token, therefore halting generation. This paper introduces an unsupervised debiasing strategy to be introduced during training in order to encourage longer sequences from base models. The experiments show that this method is effective across a range of VLMs and can be applied unsupervised with pre-training data (and therefore is somewhat task agnostic even if captioning is the submission's focus).

Reviewers praise the simple, general-purpose implementation of the method and the direct comparison between the same checkpoints trained with and without the debiasing method for direct comparisons. Reviewers also appreciate the experimental insights and the thoroughness of the investigation. The discussion period looks to have raised several areas (coherence scores, data coarseness) which can be polished in the paper even if the core components of these areas _were already in the submission_. Discussion has shown where this paper can be made clearer and this will likely be of value to the COLM program.